# Language models outperform cloze predictability in a cognitive model of reading

**Adrielli Tina Lopes Rego** [1]*, **Joshua Snell** [2], **Martijn Meeter** [1]

**1** Department of Education, Vrije Universiteit Amsterdam, and LEARN! Research Institute, Amsterdam, The Netherlands, **2** Department of Experimental and Applied Psychology, Vrije Universiteit Amsterdam, Amsterdam, The Netherlands

* a.t.lopesrego@vu.nl

**Data Availability Statement:** All the relevant data and source code used to produce the results and analyses presented in this manuscript are available on a Github repository at https://github.com/dritlopes/OB1-reader-model.

## Abstract

Although word predictability is commonly considered an important factor in reading, sophisticated accounts of predictability in theories of reading are lacking. Computational models of reading traditionally use cloze norming as a proxy of word predictability, but what cloze norms precisely capture remains unclear. This study investigates whether large language models (LLMs) can fill this gap. Contextual predictions are implemented via a novel parallel-graded mechanism, where all predicted words at a given position are pre-activated as a function of contextual certainty, which varies dynamically as text processing unfolds. Through reading simulations with OB1-reader, a cognitive model of word recognition and eye-movement control in reading, we compare the model's fit to eye-movement data when using predictability values derived from a cloze task against those derived from LLMs (GPT-2 and LLaMA). Root Mean Square Error between simulated and human eye movements indicates that LLM predictability provides a better fit than cloze. This is the first study to use LLMs to augment a cognitive model of reading with higher-order language processing while proposing a mechanism on the interplay between word predictability and eye movements.

## Author summary

Reading comprehension is a crucial skill that is highly predictive of later success in education. One aspect of efficient reading is our ability to predict what is coming next in the text based on the current context. Although we know predictions take place during reading, the mechanism through which contextual facilitation affects oculomotor behaviour in reading is not yet well-understood. Here, we model this mechanism and test different measures of predictability (computational vs. empirical) by simulating eye movements with a cognitive model of reading. Our results suggest that, when implemented with our novel mechanism, a computational measure of predictability provides better fits to eye movements in reading than a traditional empirical measure. With this model, we scrutinize how predictions about upcoming input affects eye movements in reading, and how computational approaches to measuring predictability may support theory testing. Modelling aspects of reading comprehension and testing them against human behaviour contributes to the effort of advancing theory building in reading research. In the longer term,

**Funding:** This study was supported by the Nederlandse Organisatie voor Wetenschappelijk Onderzoek (NWO) Open Competition-SSH (Social Sciences and Humanities) (https://www.nwo.nl), 406.21.GO.019 to MM. The funders had no role in study design, data collection and analysis, decision to publish, or preparation of the manuscript.

**Competing interests:** The authors declare that no competing interests exist.

more understanding of reading comprehension may help improve reading pedagogies, diagnoses and treatments.

## Introduction

Humans can read remarkably efficiently. What underlies efficient reading has been subject of considerable interest in psycholinguistic research. A prominent hypothesis is that we can generally keep up with the rapid pace of language input because language processing is predictive, i.e., as reading unfolds, the reader anticipates some information about the upcoming input [1–3]. Despite general agreement that this is the case, it remains unclear how to best operationalize contextual predictions [3,4]. In current models of reading [5–9]), the influence of prior context on word recognition is operationalized using cloze norming, which is the proportion of participants that complete a textual sequence by answering a given word. However, cloze norming has both theoretical and practical limitations, which are outlined below [4,10,11]. To address these concerns, in the present work we explore the use of Large Language Models (LLMs) as an alternative means to account for contextual predictions in computational models of reading. In the remainder of this section, we discuss the limitations of the current implementation of contextual predictions in models of reading, which includes the use of cloze norming, as well as the potential benefits of LLM outputs as a proxy of word predictability. We also offer a novel parsimonious account of how these predictions gradually unfold during text processing.

Computational models of reading are formalized theories about the cognitive mechanisms that may take place during reading. The most prominent type of model are models of eye-movement control in text reading (see [12] for a detailed overview). These attempt to explain how the brain guides the eyes, by combining perceptual, oculomotor, and linguistic processes. Despite the success of these models in simulating some word-level effects on reading behaviour, the implementation of contextual influences on the recognition of incoming linguistic input is yet largely simplified. Word predictability affects lexical access of the upcoming word by modulating either its recognition threshold (e.g. E-Z Reader [5] and OB1-reader [6]) or its activation (e.g. SWIFT [7]). This process is embedded in the "familiarity check" of the E-Z Reader model and the "rate of activation" in the SWIFT model. One common assumption among models is that the effect of predictability depends on certain word-processing stages. In the case of the E-Z Reader model, the effect of predictability of word *n* on its familiarity check depends on the completion of "lexical access" of word *n-1*. That is, predictability of word *n* facilitates its processing only if the preceding word has been correctly recognized and integrated into the current sentence representation [13]. In the case of the SWIFT model, the modulation of predictability on the rate of activation of word *n* depends on whether the processing of word *n* is on its "parafoveal preprocessing" stage, where activation increases more slowly the higher the predictability, or its "lexical completion" stage, where activation decreases more slowly the higher the predictability [12]. These models ignore the predictability of words that do not appear in the stimulus text, even though they may have been predicted at a given text position and assume a one-to-one match between the input and the actual text for computing predictability values. Because the models do not provide a deeper account of language processing at the syntactic and semantic levels, they cannot allow predictability to vary dynamically as text processing unfolds. Instead, predictability is computed prior to the simulations and fixed.

What is more, the pre-determined, fixed predictability value in such models is conventionally operationalized with cloze norming [14]. Cloze predictability is obtained by having

participants write continuations of an incomplete sequence, and then taking the proportion of participants that have answered a given word as the cloze probability of that word. The assumption is that the participants draw on their individual lexical probability distributions to fill in the blank, and that cloze reflects some overall subjective probability distribution. For example, *house* may be more probable than *place* to complete *I met him at my* for participant A, but not for participant B. However, scientists have questioned this assumption [4,10,11]. The cloze task is an offline and untimed task, leaving ample room for participants to consciously reflect on sequence completion and adopt strategic decisions [4]. This may be quite different from normal reading where only ~200ms is spent on each word [15]. Another issue is that cloze cannot provide estimates for low-probability continuations, in contrast with behavioural evidence showing predictability effects of words that never appear among cloze responses, based on other estimators, such as part-of-speech [10,16]. Thus, cloze completions likely do not perfectly match the rapid predictions that are made online as reading unfolds.

Predictability values generated by LLMs may be a suitable methodological alternative to cloze completion probabilities. LLMs are computational models whose task is to assign probabilities to sequences of words [17]. Such models are traditionally trained to accurately predict a token given its contextual sequence, similarly to a cloze task. An important difference, however, is that whereas cloze probability is an average across participants, probabilities derived from LLMs are relative to every other word in the model's vocabulary. This allows LLMs to better capture the probability of words that rarely or never appear among cloze responses, potentially revealing variation in the lower range [18]. In addition, LLMs may offer a better proxy of semantic and syntactic contextual effects, as they computationally define predictability and how it is learned from experience. The model learns lexical knowledge from the textual data, which can be seen as analogous to the language experience of humans. The meaning of words are determined by the contexts in which they appear (*distributional hypothesis* [19]) and the consolidated knowledge is used to predict the next lexical item in a sequence [11]. The advantage of language models in estimating predictability is also practical: it has been speculated that millions of samples per context would be needed in a cloze task to reach the precision of language models in reflecting language statistics [20], which is hardly feasible. And even if such an extremely large sample would be reached, we would still need the assumption that cloze-derived predictions match real-time predictions in language comprehension, which is questionable [4,21].

Importantly, language models have been shown to perform as well as, or even outperform, predictability estimates derived from cloze tasks in fitting reading data. Shain and colleagues [20] found robust word predictability effects across six corpora of eye-movements using surprisal estimates from various language models, with GPT-2 providing the best fit. The effect was qualitatively similar when using cloze estimates in the corpus for which they were available. Another compelling bit of evidence comes from Hofmann and colleagues [11], who compared cloze completion probabilities with three different language models (ngram model, recurrent neural network and topic model) in predicting eye movements. They tested the hypothesis that each language model is more suitable for capturing a different cognitive process in reading, which in turn is reflected by early versus late eye-movement measures. Item-level analyses showed that the correlations of each eye movement measure were stronger with each language model than with cloze. In addition, fixation-event based analyses revealed that the ngram model better captured lag effects on early measures (replicating the results from Smith and Levy [10]), while the recurrent neural network more consistently yielded lag effects on late measure. A more recent study [22] found neural evidence for the advantage of language models over cloze, by showing that predictions from LLMs (GPT-3, ROBERTa and ALBERT) matched N400 amplitudes more closely than cloze-derived predictions. Such evidence has led

to the belief that language models may be suitable for theory development in models of eye-movement control in reading [11].

The present study marks an important step in exploring the potential of language models in advancing our understanding of the reading brain [23,24], and more specifically, of LLMs' ability to account for contextual predictions in models of eye-movement control in reading [11]. We investigate whether a model of eye-movement control in reading can more accurately simulate reading behaviour using predictability derived from transformer-based LLMs or from cloze. We hypothesize that LLM-derived probabilities capture semantic and syntactic integration of the previous context, which in turn affects processing of upcoming bottom-up input. This effect is expected to be captured in the early reading measures (see Methods). Since predictability may also reflect semantic and syntactic integration of the predicted word with the previous context [18], late measures are also evaluated.

Importantly however, employing LLM-generated predictions is only one part of the story. A cognitive theory of reading also has to make clear how those predictions operate precisely: i.e., when, where, how, and why do predictions affect processing of upcoming text? The aforementioned models have been agnostic about this. Aiming to fill this gap, our answer, as implemented in the updated OB1-reader model, is as follows.

We propose that making predictions about upcoming words affects their recognition through graded and parallel activation. Predictability is graded because it modulates activation of all words predicted to be at a given position in the parafovea to the extent of each word's likelihood. This means that higher predictability leads to a stronger activation of all words predicted to be at a given position in the parafovea. Predictability is also parallel, because predictions can be made about multiple text positions simultaneously. Note that this is in line with the parallel structure of the OB1-reader, and this is an important contrasting point with serial processing models, such as E-Z Reader, which assume that words are processed one at a time. The predictability mechanism as proposed here is thus, in principle, not compatible with serial models of word processing. With each processing cycle, this predictability-derived activation is summed to the activity resulting from visual processing of the previous cycle and weighted by the predictability of the previous word, which in turn reflects the prediction certainty up to the current cycle (see Methods for more detailed explanation). In this way, predictability gradually and dynamically affects words in parallel, including non-text words in the model's lexicon.

Importantly, the account of predictability as predictive activation proposed here diverges from the proportional pre-activation account of predictability by Brothers and Kuperberg [25] in two ways. First, they define predictive pre-activation as the activation of linguistic features (orthographic, syntactic and semantic). However, the evidence is mixed as to whether we predict specific words [10] or more abstract categories [26]. Expectations are likely built about different levels of linguistic representations, but here predictive activation is limited to words, and this activation is roughly proportional to each word's predictability (thus we agree with the proportionality suggested by Brothers and Kuperberg [25]). Second, predictive activation would be prior to the word's availability in the bottom-up input. We note that predictability without parafoveal preview is debatable. Most studies claim that predictability effects only occur with parafoveal preview [27], but Parker and colleagues [28] showed predictability effects without parafoveal preview using a novel experimental paradigm. In OB1-reader, predictions are made about words within parafoveal preview, which means that bottom-up input is available when predictions are made about words in the parafovea. Since OB1-reader processes multiple words in parallel, predictions are generated about the identity of all words in the parafovea while their orthographic input is being processed and their recognition has not been completed.

In sum, the model assumptions regarding predictability include predictability being: (i) graded, i.e. more than one word can be predicted at each text position; (ii) parallel, i.e. predictions can be made about multiple text positions simultaneously; (iii) parafoveal, i.e. predictions are made about the words in the parafovea; (iv) dynamic, i.e. predictability effects change according to the certainty of the predictions previously made; and (v) lexical, i.e. predictions are made about words as defined by text spaces and not about other abstract categories.

Assuming that predictability during reading is graded, parallel, parafoveal, dynamic, and lexical, we hypothesize that OB1-reader achieves a better fit to human oculomotor data with LLM-derived predictions than with cloze-derived predictions. To test this hypothesis, we ran reading simulations with OB1-reader either using LLM-derived predictability or cloze-derived predictability to activate words in the model's lexicon prior to fixation. The resulting reading measures were compared with measures derived from eye-tracking data to evaluate the model's fit to human data. To our knowledge, this is the first study to combine a language model with a computational cognitive model of eye-movement control in reading to test whether the output of LLMs is a suitable proxy for word predictability in such models.

## Results

For the reading simulations, we used OB1-reader [6] (see Model Description in Methods for more details on this model). In each processing cycle from OB1-reader's reading simulation, the predictability values were used to activate the predicted words in the upcoming position (see Predictability Implementation in Methods for a detailed description). Each simulation consisted in processing all the 55 passages from the Provo Corpus [29] (see Eye-tracking and Cloze Norming in Methods for more details on the materials). The predictability values were derived from three different estimators: cloze, GPT-2 [30] and LLaMA [31]. The cloze values were taken from the Provo Corpus (see Eye-tracking and Cloze Norming in Methods for more details on Cloze Norming). The LLM values were generated from GPT-2 and LLaMA. We compare the performance of a simpler transformer-based language model, i.e. GPT-2, with a more complex one, i.e. LLaMA. Both models are transformer-based, auto-regressive LLMs, which differ in size and next-word prediction accuracy, among other aspects. The version of GPT-2 used has 124 million parameters and 50k vocabulary size. The version of LLaMA used has a much higher number of parameters, 7 billion, but a smaller vocabulary size, 32k. Importantly, LLaMA yields a higher next-word prediction accuracy on the Provo passages, 76% against 64% by GPT-2 (see Language Models in Methods for more details on these models).

We ran 100 simulations per condition in a "3x3 + 1" design: three predictability estimators (cloze, GPT-2 and LLaMA), three predictability weights (low = 0.05, medium = 0.1, and high = 0.2) and a baseline (no predictability). For the analysis, we considered eye-movement measures at word-level. The early eye-movement measures of interest were (i) skipping, i.e. the proportion of participants who skipped the word on first pass; (ii) first fixation duration, i.e. the duration of the first fixation on the word; and (iii) gaze duration, i.e. the sum of fixations on the word before the eyes move forward. The late eye-movement measures of interest were (iv) total reading time, i.e. the sum of fixation durations on the word; and (v) regression, i.e. the proportion of participants who fixated the word after the eyes have already passed the text region the word is located.

To evaluate the model simulations, we used the reading time data from the Provo corpus [29] and computed the Root Mean Squared Error (RMSE) between each eye-movement measure from each simulation by OB1-reader and each eye-movement measure from the Provo corpus averaged over participants. To check whether the simulated eye movements across predictability estimators were significantly different ($p < = .05$), we ran the Wilcoxon T-test from the Scipy python library.

In addition, we conducted a systematic analysis on the hits and failures of the simulation model across predictability conditions to better understand what drives the differences in model fit between LLMs and cloze predictability. The analysis consisted of comparing simulated eye movements and empirical eye movements on different word-based features, namely length, frequency, predictability, word id (position in the sentence), and word type (content, function, and other). In particular, word type was defined according to the word's part-of-speech tag. For instance, verbs, adjectives, and nouns were considered content words, whereas articles, particles, and pronouns were considered function words (see Reading Simulations in Methods for more details).

### Fit across eye-movement measures

In line with our hypotheses, OB1-reader simulations were closer to the human eye-movements with LLM predictability than with cloze predictability. Fig 1 shows the standardized RMSE of each condition averaged over eye movement measures and predictability weights. To attest the predictability implementation proposed in OB1-reader, we compared the RMSE scores between predictability conditions and baseline. All predictability conditions reduced the error relative to the baseline, which confirms the favourable effect of word predictability on fitting word-level eye movement measures in OB1-reader. When comparing the RMSE scores across predictability conditions, the larger language model LLaMA yielded the least error. When

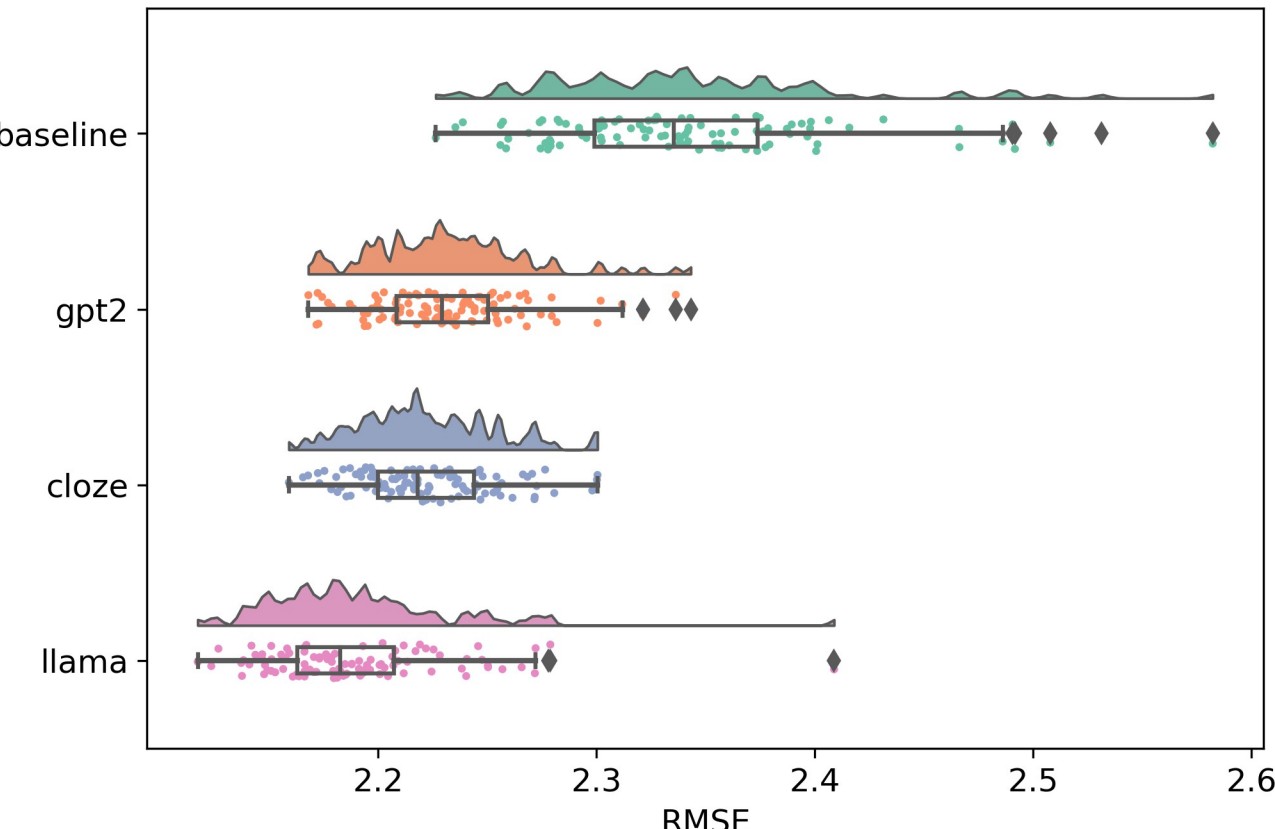

**Fig 1. Standardized RMSE scores of OB1 Reader simulations for a baseline without using word predictions, for cloze-norm predictions and predictions from the GPT-2 and LLaMA LLMs.** RMSE scores are standardized using the human averages and standard deviations as reference. The minimum RMSE value is 1, meaning no difference between eye movements from corpus and eye movements from simulations. Each data point here represents the RMSE score of one simulation averaged over words.

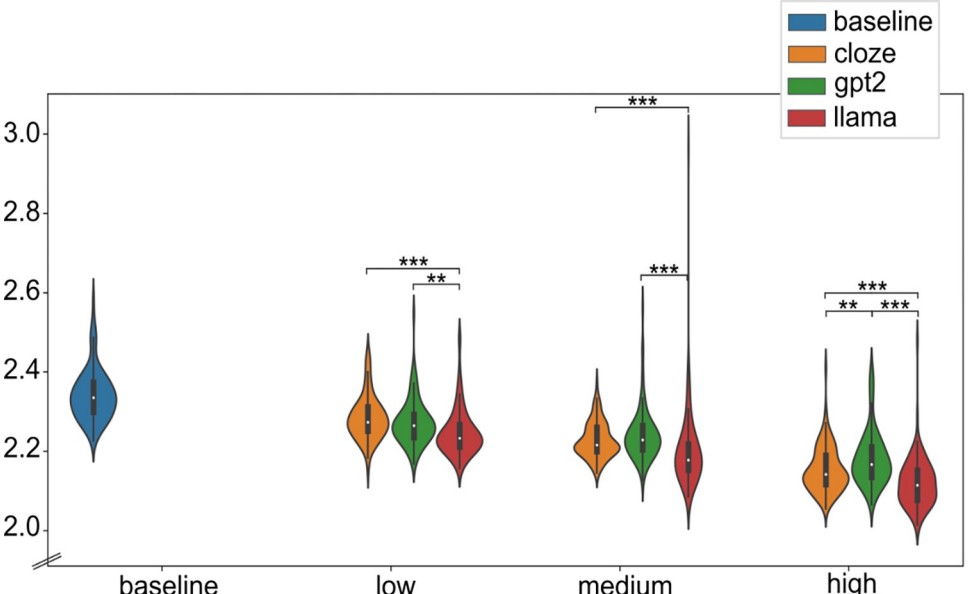

**Fig 2. Standardized RMSE scores of OB1-reader simulations per condition and predictability weight.** RMSE scores averaged over eye movement measures. * means p-value < = .05, ** means p-value < = .01, and *** means p-value < = .001.

comparing the error among predictability weights (see Fig 2), LLaMA yielded the least error in all weight conditions, while GPT-2 produced less error than cloze only with the low predictability weight. These results suggest that language models, especially with a higher parameter count and prediction accuracy, are good word predictability estimators for modelling eye movements in reading [32]. Note that the model's average word recognition accuracy was stable across predictability conditions (cloze = .91; GPT-2 = .92; LLaMA = .93). We now turn to the results for each individual type of eye movement (Fig 3).

## First fixation duration

RMSE scores for item-level first fixation duration revealed that predictability from LLaMA yielded the best fit compared to GPT-2 and cloze. LLaMA also yielded the least error in each weight condition (see Fig 4A). When comparing simulated and observed values (S1 Fig), first fixation durations are consistently longer in the model simulations. As for predictability effects (S1 Fig), the relation between predictability and first fixation duration seemed to be weakly facilitatory, with more predictability leading to slightly shorter first fixation duration in both the Provo Corpus and the OB1-reader simulations. This relation occurred in all predictability conditions, suggesting that the LLMs capture a similar relation between predictability and eye movements as cloze norming, and that this relation also exist for eye movements in the Provo Corpus.

Our systematic analysis showed that, across predictability conditions, the model generated more error with longer, infrequent, more predictable, as well as the initial words of the passage compared to the final words (S2 Fig). More importantly, the advantage of LLaMA relative to the other predictability conditions in fitting first fixation duration seems to stem from LLaMA providing better fits for highly predictable words. When comparing simulated and human first fixation durations (S3 Fig), we observed that the difference (i.e. simulated durations are longer) is more pronounced for longer and infrequent words. Another observation is that, across predictability conditions, the model fails to capture wrap-up effects (i.e. longer reaction times

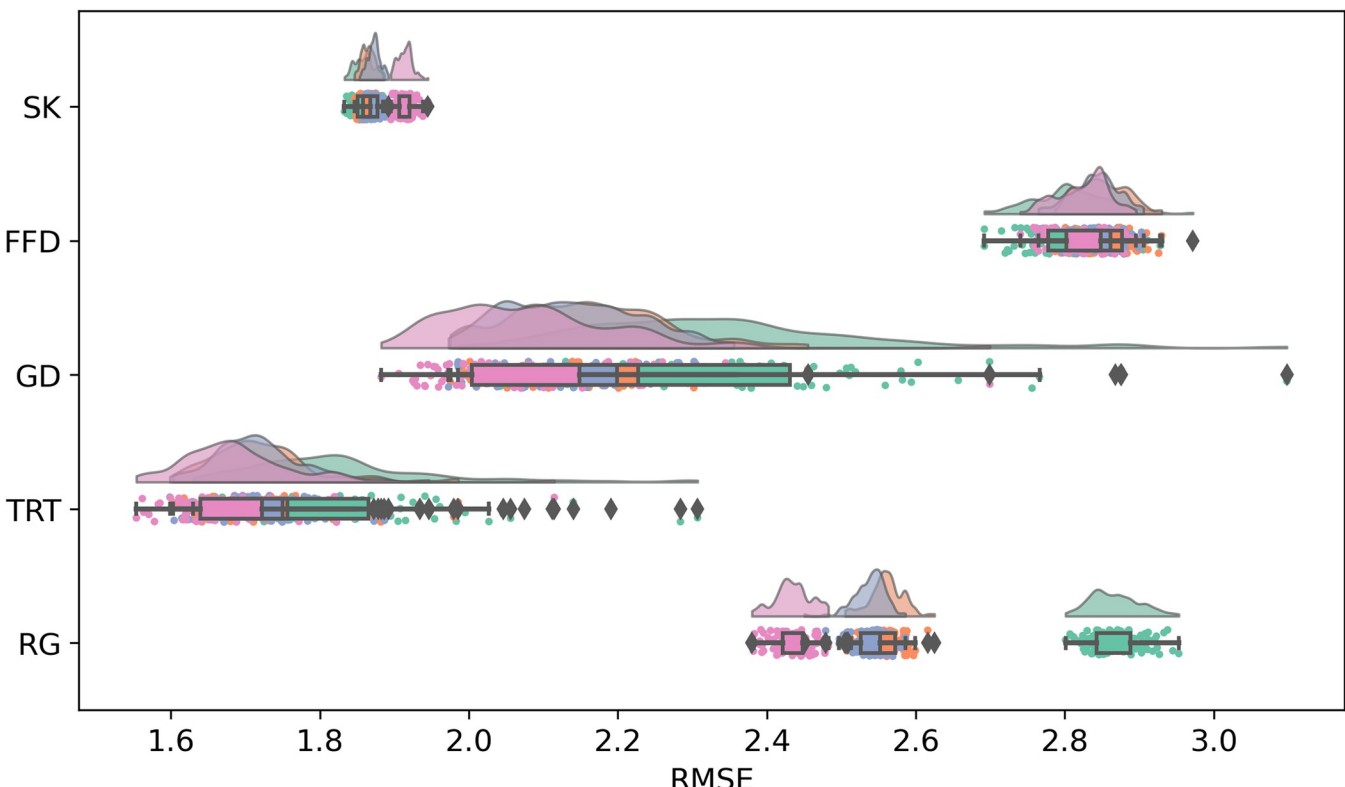

**Fig 3. Standardized RMSE scores of OB1-reader simulations per condition for each eye movement measure.** In the y-axis, eye movement measures are represented with the abbreviations SK (skipping), FFD (first fixation duration), GD (gaze duration), TRT (total reading time), and RG (regression).

towards the end of the sequence), which seems to occur in the human data, but not in the model data.

## Gaze duration

LLaMA produced the least averaged error in fitting gaze duration. GPT-2 produced either similar fits to cloze or a slightly worse fit than cloze (see Fig 4B). All predictability conditions reduce error compared to the baseline, confirming the benefit of word predictability for predicting gaze duration. Higher predictability shortened gaze duration (S4 Fig) in both the model simulations (OB1-reader) and in the empirical data (Provo Corpus), and, similarly to first fixation duration, simulated gaze durations were consistently longer than the observed gaze durations. Also consistent with first fixation durations, more error is observed for longer, infrequent words. However, differently from the pattern observed with first fixation duration, gaze durations are better fit by the model for more predictable words and initial words in a passage. LLMs, especially LLaMA, generate slightly less error across predictability values, and LLaMA is slightly better at fitting gaze durations of words positioned closer to the end of the passages (S5 Fig). Simulated values are longer than human values, especially with long words (S6 Fig).

## Skipping

Unexpectedly, skipping rates showed increasing error with predictability compared to the baseline. RMSE scores were higher in the LLaMA condition for all weights (see Fig 4C). These results show no evidence of skipping being more accurately simulated with any of the

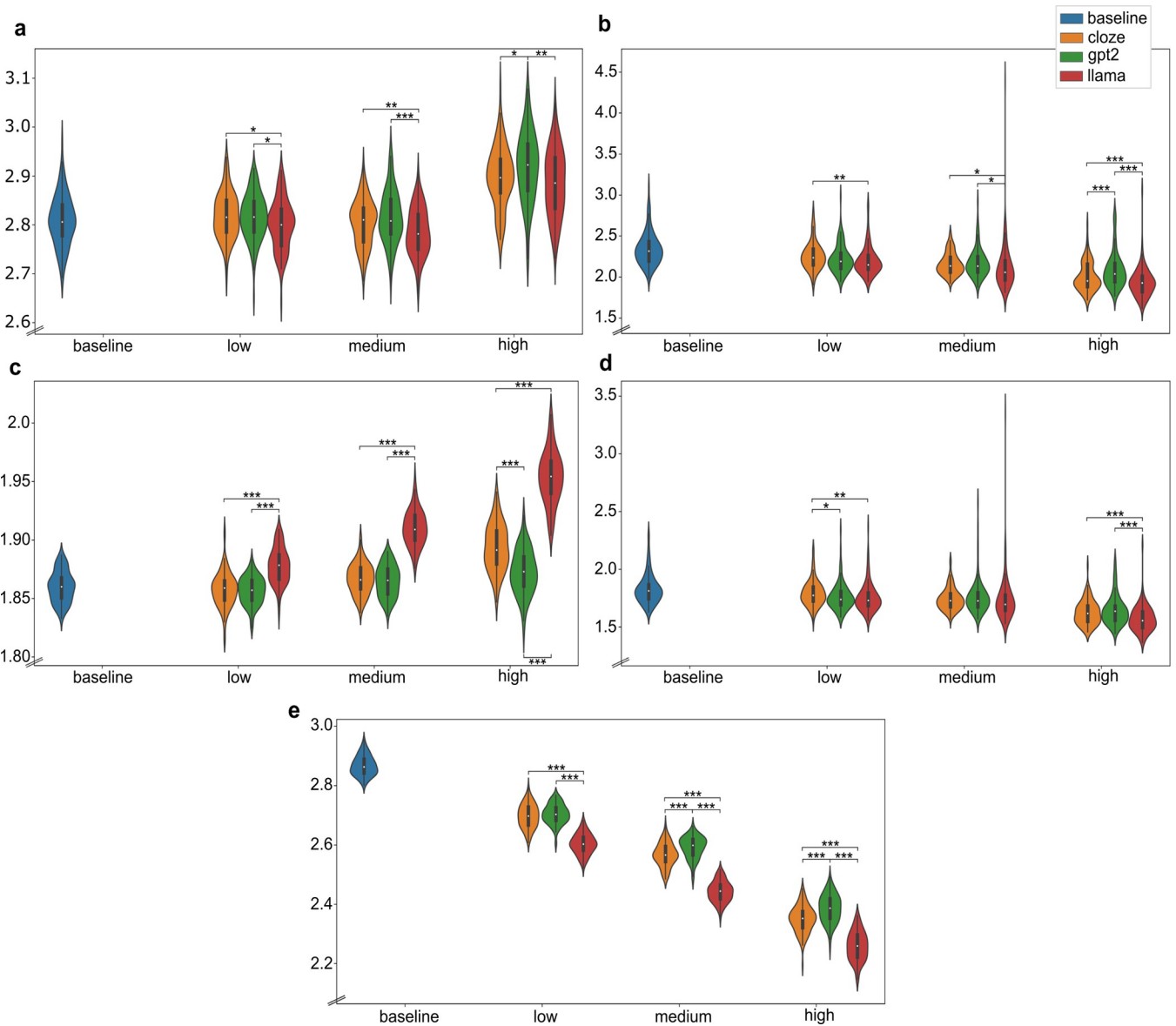

**Fig 4. Standardized RMSE scores of OB1-reader simulations per condition, eye movement measure and predictability weight.** * means p-value < = .05, ** means p-value < = .01, and *** means p-value < = .001. **a** RMSE scores for first fixation duration. **b** RMSE scores for gaze duration. **c** RMSE scores for skipping rates. **d** RMSE scores for total reading time. **e** RMSE scores for regression rates.

predictability estimations tested in this study. While OB1-reader produced sizable predictability effects on skipping rates, these effects seem to be very slight in the empirical data (S7 Fig). Another unexpected result was a general trend for producing more error for short, frequent and predictable words, with LLaMA generating more error in fitting highly predictable words than GPT-2 and cloze (S8 Fig). Moreover, the model generated more error in fitting function words than content words, which is the inverse trend relative to reading times, for which more error is observed with content words than function words (S9 Fig). A closer inspection of this pattern reveals that the model skips generally less than humans; especially longer, infrequent, and final content words. However, the reverse is seen for function words, which the model skips more often than humans do (S10 Fig).

### Total reading time

Improvement in RMSE relative to the baseline was seen for total reading time in all conditions. LLaMA showed the best performance, with lower error compared to cloze and GPT-2, especially in the low and high weight conditions (see Fig 4D). Higher predictability led to shorter total reading time, and this was reproduced by OB1 reader in all conditions. LLaMA showed trend lines for the relation between predictability and total reading time that parallel those seen in the data, suggesting a better qualitative fit than for either cloze or GPT-2 (S11 Fig). Similarly to the error patterns with gaze duration, the model generated more error with longer, infrequent, less predictable and final words across predictability conditions (S12 Fig). Also consistent with the results for gaze duration, total reading times from the simulations are longer than those of humans, particularly for longer and infrequent words (S13 Fig).

### Regression

Lastly, RMSE for regression was reduced in all predictability conditions compared to the baseline. The lowest error is generated with LLaMA-generated word predictability across predictability weights (see Fig 4E). Predictability effects on regression are in the expected direction, with higher predictability associated with lower regression rate, but this effect is amplified in the simulations (OB1-reader) relative to the empirical data (Provo Corpus) as indicated by the steeper trend lines in the simulated regression rates (S14 Fig). Similarly to the error patterns for skipping, the model generated more error with shorter, frequent and initial words. In contrast, error decreases as predictability increases in the simulations by the LLMs, especially by LLaMA, which generated less error with highly predictable words (S15 Fig). LLaMA is also slightly better at fitting regression to function words (S9 Fig). Furthermore, fitting simulated regression rates to word length, frequency and position showed similar trends as fitting the human regression rates, with steeper trend lines for the simulated values (S16 Fig).

## Discussion

The current paper is the first to reveal that large language models (LLMs) can complement cognitive models of reading at the functional level. While previous studies have shown that LLMs provide predictability estimates that can fit reading behaviour as good as or better than cloze norming, here we go a step further by showing that language models may outperform cloze norming when used in a cognitive model of reading and tested in terms of simulation fits. Our results suggest that LLMs can provide the basis for a more sophisticated account of syntactic and semantic processes in models of reading comprehension. Word predictability from language models improves fit to eye-movements in reading.

### Word predictability from language models improves fit to eye-movements in reading

Using predictability values generated from LLMs (especially LLaMA) to regulate word activation in a cognitive model of eye-movement control in reading (OB1-reader) reduced error between the simulated eye-movements and the corpus eye-movements, relative to using no predictability or to using cloze predictability. Late eye-movement measures (total reading time and regression) showed the most benefit in reducing error in the LLM predictability conditions, with decreasing error the higher the predictability weight. One interpretation of this result is that predictability reflects the ease of integrating the incoming bottom-up input with the previously processed context, with highly predictable words being more readily integrated with the previously processed context than highly unpredictable words [33]. We emphasize,

however, that we do not implement a mechanism for word integration or sentence processing in OB1-reader, and so cannot support this interpretation from the findings at that level.

Notably, the benefit of predictability is less clear for early measures (skipping, first fixation duration and gaze duration) than for late measures across conditions. The more modest beneficial effect of predictability on simulating first-pass reading may be explained by comparing simulated and observed values. We found that OB1-reader consistently provides longer first-fixation and gaze durations. Slow first-pass reading might be due to free parameters of the model not yet being optimally fit. Estimating parameters in computational cognitive models is not a simple task, particularly for models with intractable likelihood for which computing the data likelihood requires integrating over various rules. Follow-up research should apply more sophisticated techniques for model parameter fitting, for instance using Artificial Neural Networks [34].

Moreover, predictability did not improve the fit to skipping. Even though adding predictability increases the average skipping rate (which causes the model to better approximate the average skipping rate of human readers) there nonetheless appears to be a mismatch between the model and the human data in terms of which individual words are skipped. One potential explanation involves the effect of word predictability on skipping being larger in OB1-reader than in human readers. The model skips highly predictable words more often than humans do. The high skipping rate seems to be related to the relatively slow first-pass reading observed in the simulations. The longer the model spends fixating a certain word, the more activity parafoveal words may receive, and thus the higher the chance that these are recognized while in the parafovea and are subsequently skipped. This effect can be more pronounced in highly predictable words, which receive more predictive activation under parafoveal preview. Thus, given the model's assumption that early (i.e. in parafoveal preview) word recognition largely drives word skipping, predicative activation may lead OB1-reader to recognize highly predictable words in the parafovea and skip them. Parafoveal recognition either does not occur as often in humans, or does not cause human readers to skip those words as reliably as occurs in the model. It is also plausible that lexical retrieval prior to fixation is not the only factor driving skipping behaviour. More investigation is needed into the interplay between top-down feedback processes, such as predictability, and perception processes, such as visual word recognition, and the role of this interaction in saccade programming.

To better understand the potential differences between model and empirical data, as well as factors driving the higher performance of simulations using LLM-based predictability, we compared simulated and human eye movements in relation to word-based linguistic features. Across predictability conditions, we found reverse trends in simulating reading times and saccade patterns: while the reading times of longer, infrequent, and content words were more difficult for the model to simulate, more error was observed in fitting skipping and regression rates of shorter, frequent, and function words. A closer inspection of the raw differences between simulated and human data showed that the model was more critically slower than humans at first-pass reading of longer and infrequent words. It also skipped and regressed to longer, infrequent and content words less often than humans. The model, thus, seems to read more difficult words more "statically" (one-by-one) than humans do, with less room for "dynamic" reading (reading fast, skipping, and regressing for remembering or correcting).

When comparing LLMs to cloze, simulations using LLaMA-derived predictability showed an advantage at simulating gaze duration, total reading time and regression rates of highly predictable words, and a disadvantage at simulating skipping rates of highly predictable words. One potential explanation is that highly predictable words from the LLM are read faster, and thus are closer to human reading times, because LLaMA-derived likelihoods for highly predictable words are higher than those derived from GPT-2 and cloze (S17 Fig), providing more

predictive activation to those words. LLaMA is also more accurate at predicting the next word in the Provo passages, which may allow the model to provide more predictive activation to the correct word in the passage (S1 Appendix). Next to faster reading, simulations using LLaMA may also skip highly predictable words more often, leading to increasing mismatch with the human data. This process was put forward before as the reason why the model may exaggerate the skipping of highly predictable words, and, since LLaMA provides higher values for highly predictable words and it is more accurate at predicting, the process is more pronounced in the LLaMA condition.

All in all, RMSE between simulated eye movements and corpus eye movements across eye movement measures indicated that LLMs can provide word predictability estimates which are better than cloze norming at fitting eye movements with a model of reading. Moreover, the least error across eye movement simulations occurred with predictability derived from a more complex language model (in this case, LLaMA), relative to a simpler language model (GPT-2) and cloze norming. Previous studies using transformer-based language models have shown mixed evidence for a positive relation between model quality and the ability of the predictability estimates to predict human reading behaviour [20,35–37]. Our results align with studies that have found language model word prediction accuracy, commonly operationalized as perplexity or cross-entropy, and model size, commonly operationalized as the number of parameters, to positively correlate with the model's psychometric predictive power [32,35]. Note that number of parameters and next-word prediction accuracy are not the only differences between the models used. Further investigation is needed comparing more language models, and the same language model with one setting under scrutiny which varies systematically (e.g. all settings are the same except the parameter count), to help to determine which language model and settings are best for estimating predictability in reading simulations. Our results suggest that more complex pre-trained LLMs are more useful to this end.

## Language models may aid our understanding of the cognitive mechanisms underlying reading

Improved fits aside, the broader, and perhaps more important, question is whether language models may provide a better account of the higher-order cognition involved in language comprehension. Various recent studies have claimed that deep language models offer a suitable "computational framework", or "deeper explanation", for investigating the neurobiological mechanisms of language [11,23,24], based on the correlation between model performance and human data. However, correlation between model and neural and behavioural data does not necessarily mean that the model is performing cognition, because the same input-output mappings can be performed by wholly different mechanisms (this is the "multiple realizability" principle) [38]. Consequently, claiming that our results show that LLMs constitute a "deeper explanation" for predictability in reading would be a logical fallacy. It at best is a failed falsification attempt, that is, we failed to show that language models are unsuitable for complementing cognitive models of reading. Our results rather suggest that language models might be useful in the search for explanatory theories about reading. Caution remains important when drawing parallels between separate implementations, such as between language models and human cognition [39].

The question is then how we can best interpret language models for cognitive theory building. If they resemble language processing in the human brain, how so? One option is to frame LLMs as good models of how language works in the brain, which implies that LLMs and language cognition are mechanistically equivalent. This is improbable however, given that LLMs are tools built to perform language tasks efficiently, with no theoretical, empirical or biological

considerations about human cognition. It is highly unlikely that language processing in the human brain resembles a Transformer implemented on a serial processor. Indeed, some studies explicitly refrain from making such claims, despite referring to language models as a "deeper explanation" or "suitable computational framework" for understanding language cognition [11,23].

Another interpretation is that LLMs resemble the brain by performing the same task, namely to predict the upcoming linguistic input before they are perceived. Prediction as the basic mechanism underlying language is the core idea of Predictive Coding, a prominent theory in psycholinguistics [3] and in cognitive neuroscience [40,41]. However, shared tasks do not necessarily imply shared algorithms. For instance, it has been shown that more accuracy on next-word prediction was associated with worse encoding of brain responses, contrary to what the theory of predictive coding would imply [39].

Yet another possibility is that LLMs resemble human language cognition at a more abstract level: both systems encode linguistic features which are acquired through statistical learning on the basis of linguistic data. The similarities are then caused not by the algorithm, but by the features in the input which both systems learn to optimally encode. The capability of language models to encode linguistic knowledge has been taken as evidence that language—including grammar, morphology and semantics—may be acquired solely through exposure, without the need for, e.g., an in-built sense of grammar [42]. How humans acquire language has been a continuous debate between two camps: the proponents of universal grammar argue for the need of an innate, rule-based, domain-specific language system [43,44], whereas the proponents of usage-based theories emphasize the role of domain-general cognition (e.g. statistical learning, [45], and generalization, [46]) in learning from language experience. Studying large language models can only enlighten this debate if those models are taken to capture the essence of human learning from linguistic input.

In contrast, some studies criticize the use of language models to understand human processing altogether. Having found a linear relationship between predictability and reading times instead of a logarithmic relationship, Smith and Levy [10] and Brothers and Kuperberg [25] speculated the discrepancy to be due to the use of n-gram language models instead of cloze estimations. One argument was that language models and human readers are sensitive to distinct aspects of the previous linguistic context and that the interpretability and limited causal inference of language models are substantial downfalls. However, language models have become more powerful in causal inference and provide a more easily interpretable measure of predictability than does cloze. Additionally, contextualized word representations show that the previous linguistic context can be better captured by the state-of-the-art language models than by simpler architectures such as n-gram models. More importantly, neural networks allow for internal (e.g. architecture, representations) and external (e.g. input and output) probing: when certain input or architectural features can be associated with hypotheses about cognition, testing whether these features give rise to observed model behaviour can help adjudicate among different mechanistic explanations [24]. All in all, language models show good potential to be a valuable tool for investigating higher-level processing in reading. Combining language models, which are built with engineering goals in mind, with models of human cognition, might be a powerful method to test mechanistic accounts of reading comprehension. The current study is the first to apply this methodological strategy.

Finally, we emphasize that the LLM's success is likely not only a function of the LLM itself, but also of the way in which its outputs are brought to bear in the computational model. The cognitive mechanism that we proposed, in which predictions gradually affect multiple words in parallel, may align better with LLMs than with cloze norms, because the outputs of the former are based on a continuous consideration of multiple words in parallel, while the outputs

of the latter may be driven by a more serial approach. More investigation is needed as to what extent the benefit of LLMs is independent of the cognitive theory into which it is embedded. Comparing the effect of LLM-derived predictability in other models of reading, especially serial ones (e.g. E-Z Reader) could provide a clearer understanding of the generalizability of such approach.

Another potential venue for future investigations is whether transformer-based LLMs can account for early and late cognitive processes during reading by varying the size of the context window of the LLM. Hofmann et al. [11] have investigated a similar question, but using language model architectures which differ in how much of the context is considered, without including the transformer-based architecture nor performing reading simulations. We emphasize that such investigation would require careful thinking on how to align the context window of the LLM with that of the model of reading simulation. Follow-up work may address such gap.

The optimal method to investigate language comprehension may be by combining the ability of language models to functionally capture higher-order language cognition with the ability of cognitive models to mechanically capture low-order language perception. Computational models of reading, and more specifically, eye-movement control in reading, are built as a set of mathematical constructs to define and test explanatory theories or mechanism proposals regarding language processing during reading. As such, they are more interpretable and more resembling of theoretical and neurobiological accounts of cognition than LLMs. However, they often lack functional generalizability and accuracy. In contrast, large language models are built to efficiently perform natural language processing tasks, with little to no emphasizes on neurocognitive plausibility and interpretability. Interestingly, despite the reliance on performance over explanatory power and cognitive plausibility, LLMs have been shown to capture various aspects of natural language, in particular at levels of cognition considered higher order by brain and language researchers (e.g. semantics and discourse) and which models of eye-movement control in reading often lack. This remarkable ability of LLMs suggests that they offer a promising tool for expanding cognitive models of reading.

## Methods

### Eye-tracking and cloze norming

We use the full cloze completion and reading time data from the Provo corpus [29]. This corpus consists of data from 55 passages (2689 words in total) with an average of 50 words (range: 39–62) and 2.5 sentences (range: 1–5) per passage, taken from various genres, such as online news articles, popular science and fiction (see (a) below for an example passage). The Provo corpus had several advantages over other corpora; Sentences are presented as part of a multi-line passage instead of in isolation [47], which is closer to natural, continuous reading. In addition, Provo provides predictability norms for each word in the text, instead of only the final word [48], which is ideal for studies in which every word is examined. Finally, other cloze corpora tend to contain quite many constrained contexts (which are actually rare in natural reading), while this corpus provides a more naturalistic cloze probability distribution [29].

a.  There are now rumblings that Apple might soon invade the smart watch space, though the company is maintaining its customary silence. The watch doesn't have a microphone or speaker, but you can use it to control the music on your phone. You can glance at the watch face to view the artist and title of a song.

In an online survey, 470 participants provided a cloze completion to each word position in each passage. Each participant was randomly assigned to complete 5 passages, resulting in 15

unique continuations filled in by 40 participants on average. All participants were English native speakers, ages 18–50, with at least some college experience. Another 85 native English-speaking university students read the same 55 passages while their eyes were tracked with a high-resolution, EyeLink 1000 eye-tracker.

The cloze probability of each continuation in the upcoming word position was equivalent to the proportion of participants that provided the continuation in the corresponding word position. Since the number of participants completing a sequence was a maximum of 43, the minimum cloze probability of a continuation was 0.023 (i.e. if each participant would give a different continuation). Words in a passage which did not appear among the responded continuations received cloze probability of 0. The cloze probabilities of each word in each passage and the corresponding continuations were used in the model to pre-activate each predicted word, as further explained in the sub-section "Predictability Implementation" under Methods.

The main measure of interest in this study is eye movements. During reading, our eyes make continuous and rapid movements, called *saccades*, separated by pauses in which the eyes remain stationary, called *fixations*. Reading in English consists of fixations of about 250ms on average, whereas a saccade typically lasts 15-40ms. Typically, about 10–15% are saccades to earlier parts of the text, called *regressions*, and about two thirds of the saccades skip words [49].

The time spent reading a word is associated with the ease of recognizing the word and integrating it with the previously read parts of the text [49]. The fixation durations and saccade origins and destinations are commonly used to compute word-based measures that reflect how long and how often each word was fixated. Measures that reflect early stages of word processing such as lexical retrieval include (i) skipping rate, (ii) first fixation duration (the duration of the first fixation on a word), and (iii) gaze duration (the sum of fixations on a word before the eyes move forward). Late measures include total reading time (iv) and (v) regression rate and are said to reflect full syntactic and semantic integration [50]. Facilitatory effects of word predictability are generally evidenced in both early measures and late measures: that is, predictable words are skipped more often and read more quickly [27].

The measures of interest readily provided in the eye-tracking portion of the corpus were first fixation duration, gaze duration, total reading time, skipping likelihood and regression likelihood. Those measures were reported by the authors to be predictable from the cloze probabilities, attesting the validity of the data collected. We refer the reader to [29] for more details on the corpus used.

## Language models

Language model probabilities were obtained from two transformer-based language models: the smallest version available of the pre-trained LLaMA [31] (7 billion parameters, 32 hidden layers, 32 attention heads, 4096 hidden dimensions and 32k vocabulary size); and the smallest version of the pre-trained GPT-2 [30] (124 million parameters, 12 hidden layers, 12 attention heads, 768 hidden dimensions and 50k vocabulary size). Both models were freely accessible through the Hugging Face Transformers library at the time of the study. The models are auto regressive and thus trained on predicting a word based uniquely on its previous context. Given a sequence as input, the language model computes the likelihood of each word in the model's vocabulary to follow the input sequence. The likelihood values are expressed in the form of logits in the model's output vector, where each dimension contains the logit of a corresponding token in the model's vocabulary. The logits are normalized using softmax operation to be between 0 and 1.

Since the language model outputs a likelihood for each token in the model's vocabulary, we limited the sample to only the tokens with likelihood above a threshold. The threshold was

defined according to two criteria: the number of predicted words by the language model should be close to the average number of cloze responses over text positions, and the threshold value should be sufficiently low in order to capture the usually lower probabilities of language models. We have attempted a few threshold values (low = 0.001, medium-low = 0.005, medium = 0.01, medium-high = 0.05, and high = 0.1). The medium threshold (0.01) provided the closest number of continuations and average top-1 predictability estimate to those of cloze. For example, using the medium threshold on GPT-2 predictions provided an average of approximately 10 continuations (range 0–36) and an average predictability of 0.29 for the top-1 prediction, which was the closest to the values from cloze (average of 15 continuations ranging from 0 to 38, and top-1 average predictability of 0.32). Low and medium-low provided a much higher number of continuations (averages of 75 and 19, ranging up to 201 and 61, respectively), whereas medium-high and high provided too few continuations compared to cloze (average of 2 and 1 continuations, ranging up to 12 and 5, respectively). The medium threshold was also optimal for LLaMA. Note that we have not applied softmax to the resulting sequence of predictability estimates. The highly long tail of predictability estimates excluded (approximately the size of the LLM vocabulary, i.e. ~50k estimates for GPT-2 and ~32k for LLaMA) meant that re-normalizing the top 10 to 15 estimates (the average number of continuations post threshold filtering) would remove most of the variation among the top estimates. For instance, in the first passage the second word position led to likelihoods for the top 12 predictions varying between .056 and .011. Applying softmax resulted in all estimates transformed to .08 when rounded to two decimals. Thus, even though it is common practice to re-normalize after filtering to ensure the values sum to one across different sources, we opted to use predictability estimates without post-filtering re-normalization.

Each sequence was tokenized with the corresponding model's tokenizer before given as input, since the language models have their own tokenization (Byte-Pair Encoder [51]) and expect the input to be tokenized accordingly. Pre-processing the tokens and applying the threshold on the predictability values resulted in an average of 10 continuations per word position (range 1 to 26) with LLaMA and an average of 10 continuations per word position (range 1 to 36) with GPT-2. After tokenization, no trial (i.e. Provo passage) was longer than the maximum lengths allowed by LLaMA (2048 tokens) nor by GPT-2 (1024 tokens). Note that the tokens forming the language model's vocabulary do not always match a word in OB1-reader's vocabulary. This is because words can be represented as multi-tokens in the language model vocabulary. Additionally, OB1-reader's vocabulary is pre-processed and limited to the text words plus the most frequent words in a frequency corpus [52]. 31% of the predicted tokens by LLaMA were not in OB1-reader's vocabulary and 17% of words in the stimuli are split into multi-tokens in LLaMA's vocabulary. With GPT-2, these percentages were 26% and 16%, respectively.

To minimize the impact of vocabulary misalignment, we considered a match between a word in the OB1-reader's lexicon and a predicted token when the predicted token corresponded to the first token of the word as tokenized by the language model tokenizer. For instance, the word "customary" is split into the tokens "custom" and "ary" by LLaMA. If "custom" is among the top predictions from LLaMA, we used the predictability of "custom" as an estimate for the predictability of "customary". We are aware that this design choice may overestimate the predictability of long words, as well as create discrepancies between different sources of predictability (as different language models have different tokenizers). However, in the proposed predictability mechanism, not only the text words are considered, but also all the words predicted at a given text position (above a pre-defined threshold). Other approaches that aggregate the predictability estimates over all tokens belonging to the word, instead of only the first token, would require computing next-token predictions repeatedly for each

different predicted token for each text position, until we assume a word has been formed. To avoid these issues and since the proportion of text words which are split into more than one token by the language models is moderate (17% and 16% from LLaMA and GPT-2, respectively), we adopted the simpler first-token-only strategy.

## Model description

In each fixation by OB1-reader (illustrated in Fig 5), the input consists of the fixated word $n$ and the words $n-1$, $n + 1$, $n + 2$ and $n + 3$ processed in parallel. With each processing cycle, word activation is determined by excitation from constituent letters, inhibition from competing words and a passive decay over time.

The model assumes that attention is allocated to multiple words in parallel, such that recognizing those words can be achieved concurrently. Open bigrams [53] from three to five words are activated simultaneously, modulated by the eccentricity of each letter, its distance from the

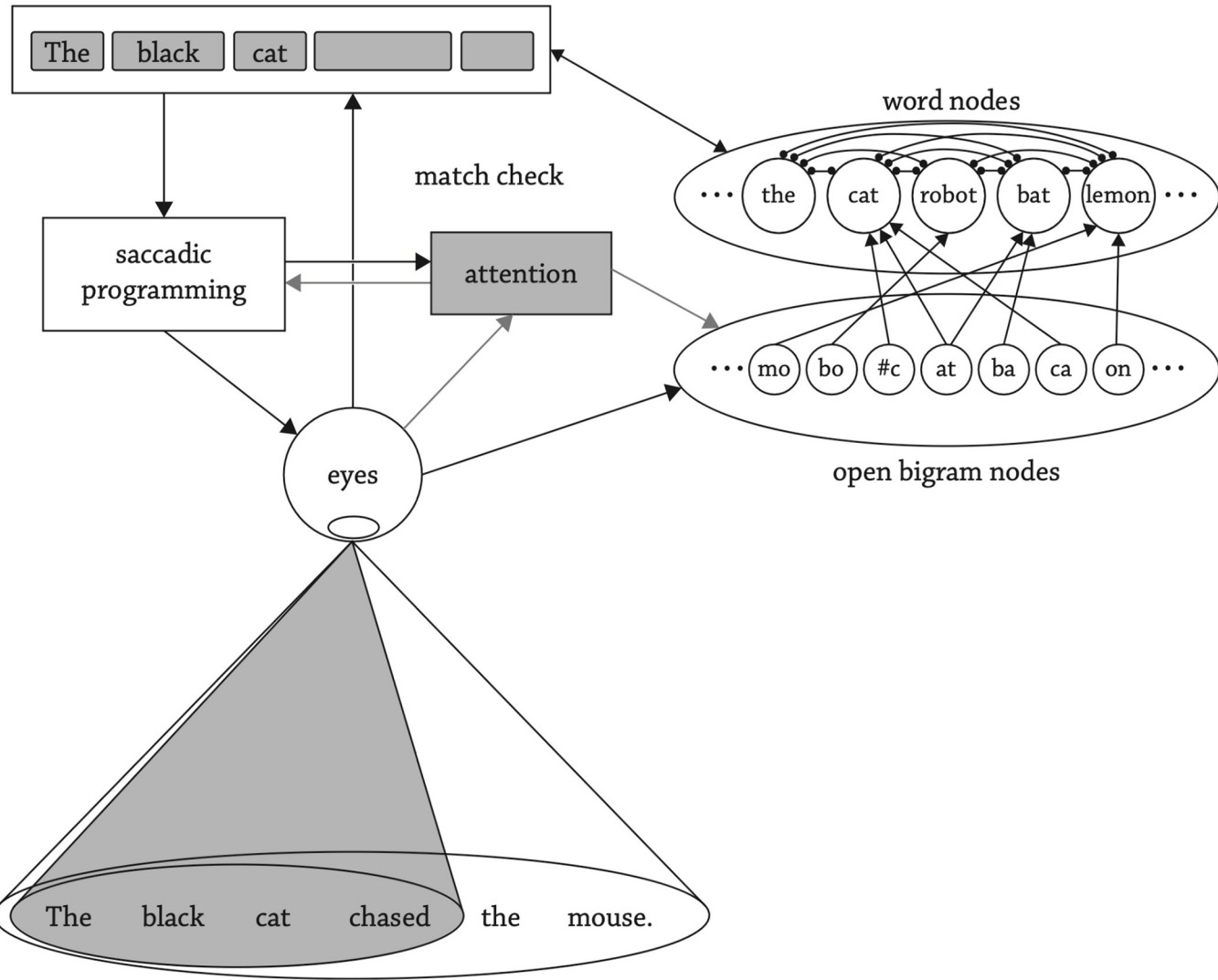

**Fig 5. Schematic Diagram from OB1-reader.** This diagram was taken from [12].

focus of attention, and crowding exerted by its neighbouring letters. Each fixation sets off several word activation cycles of 25ms each. Within each cycle, the bigrams encoded from the visual input propagate activation to each lexicon word they are in. The activated words with bigram overlap inhibit each other. Lexical retrieval only occurs when a word of similar length to the word in the visual input reaches the recognition threshold, which depends on its frequency. Initiating a saccade is a stochastic decision, with successful word recognition increasing the likelihood of moving the "eyes", that is, the letter position where the fixation point is simulated in the text. Word recognition also influences the size of the attention window, by increasing the attention window when successful. Once the saccade is initiated, the most visually salient word within the attention window becomes the saccade's destination. With the change in the location of fixation, the activation of words no longer in the visual input decays, while words encoded in the new visual input receive activation.

The strength of the excitation from visual processing $v_i$ generated by a letter $i$ depends on its eccentricity $e_i$ and crowding $m_i$, weighted by an attentional weight $a_i$, and normalized by the number of bigrams in the lexicon $len(bigrams)$ and a constant $c_d$, as in the following equation (adapted from Snell et al. [6]):

$$v_i = \frac{a_i \times m_i \left[ 1/c_e \left( 0.018 * e_i + \frac{1}{0.64} \right) \right]}{len(bigrams) + c_d}$$

The combinations of letters activate open bigrams, which are pair of letters within the same word and, in OB1-reader, up to three letters apart. The activation of an open bigram $O_{ij}$ thus equals the square root of the visual input $v_i$ and $v_j$ of the constituent letters $i$ and $j$, as implemented in the original OB1-reader.

The strength of the inhibition from other word nodes depends on the extent of the pairwise orthographic overlap. With each time step that the word is not in visual input, the activation decays. Words are recognized when their activations reach their corresponding recognition thresholds. Note that in the original OB1-reader, this threshold is determined by the word's length, frequency, and predictability, which is constant for all words being predicted at the same text position. In this version of OB1-reader, predictability is modelled as activation instead. This allows predictability to be specific to the words being predicted beyond the actual word in the text position. The recognition threshold is thus only determined by frequency weighted by a constant ($c_f$), as follows:

$$T = freq_{max} * \frac{(freq_{max}/c_f) - freq_w}{freq_{max}/c_f}$$

## Predictability implementation

In addition to the visual activation, words predicted to follow the word being currently read receive predictive activation ($A_p$) in the model's lexicon. Given a word $w_i$ at position i in a passage T, activation is added to each predicted word $w_p$ of predictability $pred_w$ above threshold t in the model's lexicon, prior to word $w_i$ being fixated by the model. The activation $A_p$ of $w_p$ as a result of $w_p$ being predicted from the previous context is defined as follows:

$$A_p = pred_w \times pred_{w-1} \times c_p,$$

where $pred_w$ is either the language model or cloze completion probability; $pred_{w-1}$ is the predictability of the previous word, which varies as a function of recognition of that word; and $c_1$ is a free-scaling parameter. $pred_{w-1}$ equals to the cloze- or language model-derived predictability of the previous word if it has not been recognized yet, or 1 if its lexical access has been

**Table 1. Simulation Parameters.**

| Parameter | Description | Value |
|---|---|---|
| $c_e$ | Scaling cortical magnification | 35.55556 |
| $e_i$ | Letter eccentricity | Distance in letters between letter and centre of attention, times 0.3 (letter size per degree of visual angle) |
| $m_i$ | Masking factor describing crowding | 1 for outer letters, .5 for inner letters, 3 for one-letter words |
| $c_d$ | Discounted ngram factor to normalize strong activation of short words | 5 |
| $c_f$ | Weight of frequency in threshold setting | .08 |
| $c_p$ | Weight of predictability in predictive activation setting | [.05, .1, .2] |
| $\tau$ | Decay | -.1 |
| $S_{max}$ | Maximum activity of a word node | 1 |
| $S_{min}$ | Minimum activity of a word node | 0 |
| $c_1$ | Bigram-to-word excitation | 1 |
| $c_2$ | Word-to-word inhibition | -2.5 |

completed.

$$\Delta S_w = ((S_{max} - S_w) \times [c_1(\Sigma_{ij\epsilon w}O_{ij}) - c_2(\Sigma_k d_{w,k}S_k) + A_p]) + ((S_w - S_{min})*\tau)$$

The input between square brackets has three parts. The first term ($c_1(\Sigma_{ij\epsilon w}O_{ij})$) is the excitation from bigram nodes, the second term ($-c_2(\Sigma_k d_{w,k}S_k)$) is the inhibition exerted by competing words in the lexicon, and the third part ($A_p$) is the predictive activation. Word activity is bound to the interval between $S_{max}$ and $S_{min}$ and decays ($\tau$) with every cycle in which the word is not in the visual input. Each parafoveal word gets predictive activation in parallel to other parafoveal words until its recognition cycle is reached. See Table 1 for an overview of the model parameters.

With the described implementation, predictive activation exerts a facilitatory effect on word recognition through faster reading times and more likelihood of skipping. Words in the text which are predicted receive additional activation prior to their fixation, which allows for the activation threshold for lexical access to be reached more easily. Consequently, the predicted word may be recognized more readily and even before fixation. In addition, higher predictability may indirectly increase the likelihood of skipping, because more successful recognition in the model leads to a larger attention window. In contrast, activation of predicted words may exert an inhibitory effect on the recognition of words with low predictability, because activated, predicted words inhibit the words that they resemble orthographically, which may include the word that was truly in the text.

## Reading simulations

The simulation input consisted of each of the 55 passages from the Provo corpus, with punctuation and trailing spaces removed and words lower-cased. The output of each model simulation mainly consisted of reading times, skipping and regression computed per simulated fixation. For evaluation, we transform the fixation-centered output into word-centered data, where each data point aggregates fixation information of each word in each passage. Using the word-centered data, we computed the eye-movement measures for each word in the corpus. Since the eye movement measures vary on scale (milliseconds for durations and likelihood for skipping and regression), we standardized the raw differences between simulated and human

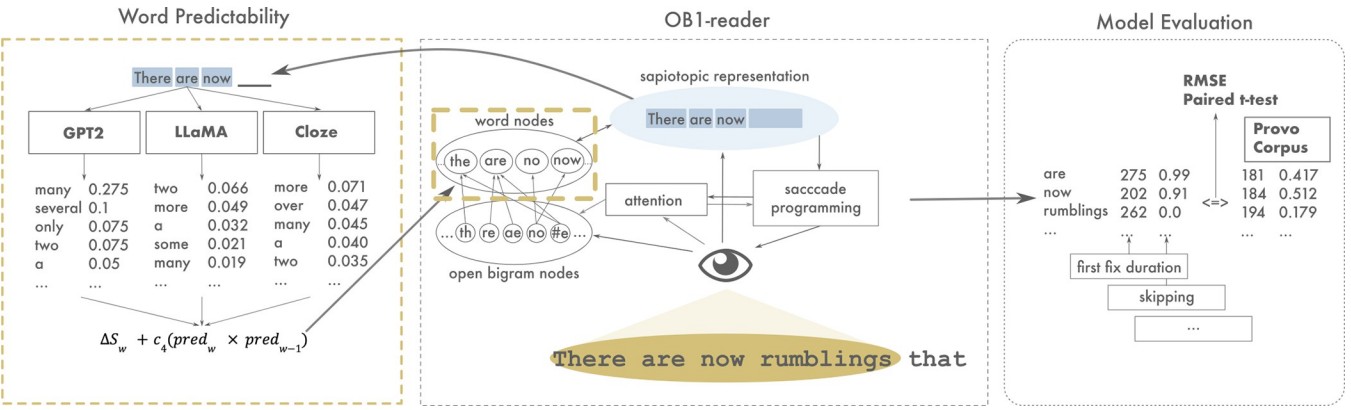

**Fig 6. Methodology used in experiment with OB1-reader, LLMs and Cloze Predictability.** (a) In OB1-reader, a model of eye-movements in reading, word predictability is computed for every word predicted in a condition (GPT-2, LLaMA, cloze) in each position in the current stimulus. The predictability of each predicted word (*pred*w) is weighted by the predictability of the previous word (*pred*w-1) and a free parameter (*c*4), and added to its current activation (*S*w) at each cycle until recognition. The number of processing cycles by OB1-reader to achieve recognition determine the word's reading time. (b) Eye movements simulated by the model are compared to the eye movements from the Provo Corpus by computing RMSE scores.

values based on the respective human average. Root Mean Square Error (RMSE) was then calculated between the simulated word-based eye-movement measures and the equivalent human data per simulation, and Wilcoxon t-test was run to compare the RMSE across conditions.

Finally, we compared the error in each condition across different word-based linguistic variables (length, frequency, predictability, part-of-speech category and position) to better understand the differences in performance. This analysis consisted of binning (20 bins of equal width) the continuous linguistic variables (length, frequency, predictability and position), computing the RMSE in each simulation for each bin, and averaging the RMSE for each bin over simulations. Part-of-speech tags were binned into three categories: content, consisting of the Spacy part-of-speech tags noun (NOUN), verb (VERB), adjective (ADJ), adverb (ADV) and proper noun (PROPN); function, consisting of the Spacy part-of-speech tags auxiliary (AUX), adposition (ADP), conjunction (CONJ, SCONJ, CCONJ), determiner (DET), particle (PART), and pronoun (PRON); and other, consisting of numeral (NUM), interjection (INTJ) and miscellaneous (X). See Fig 6 for an overview of the methodology. The code to run and evaluate the model simulations is fully available on the GitHub repository of this project.

## Supporting information

**S1 Fig. Relation between predictability and first fixation duration.** Predictability is computed from cloze norms, gpt2 or llama, and is displayed as a function of the weight attached to predictability in word activation (low, medium or high). First fixation duration is displayed in milliseconds.
(TIFF)

**S2 Fig. Root Mean Square Error (RMSE) for first fixation durations in relation to word variables.** The word variables are frequency, length, predictability, and the position of the word in the passage (word_id).
(TIFF)

**S3 Fig. Relation between word variables and first fixation duration.** The word variables are frequency, length, predictability, the position of the word in the passage (word_id), and type

(function, content, or other). First fixation duration is displayed in milliseconds.
(TIFF)

**S4 Fig. Relation between predictability and gaze duration.** Predictability is computed from cloze norms, gpt2 or llama, and is displayed as a function of the weight attached to predictability in word activation (low, medium or high). Gaze duration is displayed in milliseconds.
(TIFF)

**S5 Fig. Root Mean Square Error (RMSE) for gaze durations in relation to word variables.** The word variables are frequency, length, predictability, and the position of the word in the passage (word_id).
(TIFF)

**S6 Fig. Relation between word variables and gaze duration.** The word variables are frequency, length, predictability, the position of the word in the passage (word_id), and type (function, content, or other). Gaze duration is displayed in milliseconds.
(TIFF)

**S7 Fig. Relation between predictability values and skipping likelihood.** Predictability is computed from cloze norms, gpt2 or llama, and is displayed as a function of the weight attached to predictability in word activation (low, medium or high).
(TIFF)

**S8 Fig. Root Mean Square Error (RMSE) for skipping likelihood in relation to word variables.** The word variables are frequency, length, predictability, and the position of the word in the passage (word_id).
(TIFF)

**S9 Fig. Root Mean Square Error (RMSE) for each eye movement measure in relation to word type (content, function or other).**
(TIFF)

**S10 Fig. Relation between word variables and skipping likelihood.** The word variables are frequency, length, predictability, the position of the word in the passage (word_id), and type (function, content, or other).
(TIFF)

**S11 Fig. Relation between predictability and total reading time.** Predictability is computed from cloze norms, gpt2 or llama, and is displayed as a function of the weight attached to predictability in word activation (low, medium or high). Total reading time is displayed in milliseconds.
(TIFF)

**S12 Fig. Root Mean Square Error (RMSE) for total reading time in relation to word variables.** The word variables are frequency, length, predictability, and the position of the word in the passage (word_id).
(TIFF)

**S13 Fig. Relation between word variables and total reading time.** The word variables are frequency, length, predictability, the position of the word in the passage (word_id), and type (function, content, or other). Total reading time is displayed in milliseconds.
(TIFF)

**S14 Fig. Relation between predictability and regression likelihood.** Predictability is computed from cloze norms, gpt2 or llama, and is displayed as a function of the weight attached to predictability in word activation (low, medium or high).
(TIFF)

**S15 Fig. Root Mean Square Error (RMSE) for regression likelihood in relation to word variables.** The word variables are frequency, length, predictability, and the position of the word in the passage (word_id).
(TIFF)

**S16 Fig. Relation between word variables and regression likelihood.** The word variables are frequency, length, predictability, the position of the word in the passage (word_id), and type (function, content, or other).
(TIFF)

**S17 Fig. Distribution of predictability values by each predictor.**
(TIFF)

**S1 Appendix. Accuracy and Correlation Analyses.**
(DOCX)

## Author Contributions

**Conceptualization:** Adrielli Tina Lopes Rego, Joshua Snell, Martijn Meeter.

**Data curation:** Adrielli Tina Lopes Rego.

**Formal analysis:** Adrielli Tina Lopes Rego, Martijn Meeter.

**Funding acquisition:** Martijn Meeter.

**Investigation:** Adrielli Tina Lopes Rego.

**Methodology:** Adrielli Tina Lopes Rego, Joshua Snell, Martijn Meeter.

**Project administration:** Martijn Meeter.

**Software:** Adrielli Tina Lopes Rego, Martijn Meeter.

**Supervision:** Joshua Snell.

**Validation:** Martijn Meeter.

**Visualization:** Adrielli Tina Lopes Rego.

**Writing – original draft:** Adrielli Tina Lopes Rego.

**Writing – review & editing:** Adrielli Tina Lopes Rego, Joshua Snell, Martijn Meeter.

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
