## [Decision Letter · Decision Letter 0]

10 Jun 2024

Dear Ms. Lopes Rego,

Thank you very much for submitting your manuscript "Language models outperform cloze predictability in a cognitive model of reading" for consideration at PLOS Computational Biology.

As with all papers reviewed by the journal, your manuscript was reviewed by members of the editorial board and by several independent reviewers (three in this case). In light of the reviews (below this email), we would like to invite the resubmission of a significantly-revised version that takes into account the reviewers' comments.

One point that was brought up by two of the reviewers – and which I also scribbled down in my own evaluation of your work – is that the methodology should be described in more detail. In addition to the examples of missing details raised by the reviewer, another one is that it was unclear to me what the input to each simulation was; apparently 55 passages from a corpus were used, but how long were those passages, what did they look like, and how were they chosen?

A second point that was brought up by two reviewers and also appeared in my own notes was that the manuscript would benefit from a discussion of systematic differences between model fit and empirical data. Would it be possible to say something about why the LLM-based measure leads to lower RMSE? On what kinds of inputs does it perform particularly well and are there cases where it does substantially worse? Answers to those kinds of questions (and those raised by the reviewers) could give important insights that go beyond just comparing performances.

Related to this, a third important point that was brought up is the question of whether the success of the LLM is purely due to its scale (trained on texts from millions of people vs. the 40 participants from which the empirical Cloze norms were derived)?

Besides these three points, the reviewers have numerous other small comments and suggestions. You can find their detailed reviews below.

We cannot make any decision about publication until we have seen the revised manuscript and your response to the reviewers' comments. Your revised manuscript will be sent to the three reviewers for further evaluation.

Sincerely,

Ronald van den Berg

Academic Editor

PLOS Computational Biology

Daniele Marinazzo

Section Editor

PLOS Computational Biology

Reviewer's Responses to Questions

**Comments to the Authors:**

Reviewer #1: Review of manuscript entitled “Language models outperform cloze predictability in a cognitive model of reading” (PCOMPBIOL-D-24-00700) by Rego, Snell, and Meeter.

Summary. The manuscript describes a simulation “experiment” in which OB1-Reader was used to predict the eye movements observed during the reading of the Provo corpus. Across the simulated “conditions,” the predictability of each word in the corpus was set equal to zero or values derived from cloze norms or one of two large language models, LLMs (GPT2 or LLaMA), with the weights being used by OB1-Reader to modulate the effect of predictability also being manipulated. The key result from this exercise was that predictability values derived from LLMs provided better fits to the data than the cloze norms. The authors discuss their work in terms of possible theoretical interpretations of LLMs and their integration with more plausible models of reading.

General comments. I enjoyed reading this manuscript and thought that the basic idea of using LLMs to generate word-predictability values for reading models was both novel and interesting. For those reasons, I think the manuscript is potentially worthy of publication. Before that can happen, however, the manuscript will require a significant amount of revision to address the concerns outlined below.

Major Concerns/Questions:

1. Many of the essential details of OB1-Reader and how the simulations were completed were not provided, making it difficult to evaluate the quality of the simulation results. For example, the authors first mention “pre-cycle activation” on p. 7 but only provide some of the details to understand what this is later, on pp. 24-25. Also, although the authors provide a figure of their model, they do not provide any of the implementational details, but instead assume that readers will consult Snell et al. (2018). The action editor may differ with me here, but my view is that this type of paper should be self-contained, and that whatever model details are required to understand the simulations should be provided.

In relation to this, one important detail that needs to be expanded upon is related to the issue of serial versus parallel processing of words. For example, on p. 25, the authors indicate that predictability is “parallel, i.e., predictions can be made about multiple text positions simultaneously.” This assumption obviously aligns with the architecture of OB1-Reader but is contrary to models that, like E-Z Reader, assume that words are processed one at a time. In fact, in E-Z Reader, a word’s predictability can only be used to facilitate its processing if the preceding words have been both identified and integrated into the sentence representation being constructed (see Reichle et al., 2012, p. 159). This point of contrast should be mentioned because the two models, OB1-Reader and E-Z Reader, represent the end points on an important theoretical continuum.

2. On pp. 12-14 and in the Supporting Information, the authors discuss the model fits and how they compare to the observed data from the Provo corpus. However, the authors do not discuss the systematic differences between the two. For example, in Figures 1 and 2 of the Supporting Information, the simulated first fixation and gaze durations are consistently longer than the corresponding observed values. Some explanation of these differences seems necessary.

3. I’m not an expert on LLMs, but my understanding is that part of their utility reflects the fact that they’ve been trained on huge volumes of text, perhaps making their “experience” (using the authors’ term; see p. 6) equivalent to that of many readers (as opposed to, e.g., what would be expected of a single reader). In contrast, the empirical cloze norms taken from the Provo corpus were based on an average of only 40 participants (p. 19). If that’s the case, would the model’s performance using cloze norms be more equivalent to the model’s performance using LMMs if the norms had been based on a much larger sample? Some explanation seems necessary. And if my intuition here is incorrect, then a discussion of precisely why might strengthen the authors’ argument for the value of using LMMs.

Minor Concerns: These suggestions are admittedly nitpicky but intended to be helpful [page/paragraph/line(s)].

1. General comment: I had the impression that the manuscript was written in haste, with minimal effort to proof it for typos.

2. 2 (abstract)/1/2: Remove “yet”.

3. 3/1/2: “provide”  “provides”

4. 3/1/penultimate sentence: The sentence beginning “In the short term,” sound empty and should be expanded or removed.

5. 4/1/6: Add two references: (1) Reilly & Radach (2006); and (2) Li & Pollatsek (2020).

6. 4/1/7 (and elsewhere throughout manuscript): “Cloze” is typically not capitalized.

7. 4/3/4 (and elsewhere): Use the Oxford comma with lists of three or more items to avoid ambiguity.

8. 5/1/2: “EZ-reader”  “E-Z Reader”

9. 5/1/final sentence: Although what is said here is technically true, in E-Z Reader, the cloze values are “fixed” only because the model doesn’t provide a deep account of language processing. As such, I don’t conceptualize cloze values as being static (as suggested by the authors) but instead view this assumption as a proxy. Some acknowledgment of this seems necessary. (And not just for my model, but for the others, too.)

10. 5/2/6-7: Move the parenthetical example to the end, as a new sentence.

11. 6/1/7: “is determined”  “are determined”

12. 6/1/10: “the phrase “as well as (17), or outperform (18)” should be re-written and expanded, with more details provided (i.e., not just the citations).

13. 6/2/2: Insert “our” after “advancing”.

14. 8/2: I would remove the abbreviations for the eye-movement measures because they are not used elsewhere in the manuscript (except Figure 2, where they are provided). Also, I would list the measures with numbers; e.g., “… of interest were: (1) skipping; i.e., …” Do the same on p. 20.

15. p. 12, Figure 3: The axes labels are too small to read. Also, the p-value associated with ** is not indicated in the note.

16. 14/3/4: “as-”  “as”

17. 14/3/7: Shouldn’t this read “semantic and syntactic processes” given the discussion on p. 6?

18. 15/2/5: Remove “specifically”.

19. 15/2/6: “higher”  “larger”

20. 15/2/7-11: The argument beginning with “Given the model’s assumption…” doesn’t make sense to me. I would suggest clarifying this argument or removing it.

21. 16/1/6: Shouldn’t “model quality” be “model size”?

22. 16/2/11: I think I understood what the authors meant by “failed falsification attempt” but I suspect that many readers won’t. So please expand this point.

23. 17/3: In the preceding paragraphs, the authors briefly describe how LLMs have been interpreted and then provide an argument for why each of these interpretations is limited. The authors don’t do this for the interpretation mentioned in this paragraph, however. That said, is there any evidence supporting or refuting this interpretation?

24. 18/1/2: Remove “and more” and “still”.

25. 18/1/7: Again, I think I understand what the authors mean by “probing” the internal versus external behavior of neural networks, but I suspect that most readers won’t. So briefly expand this point.

26. 19/1/6: The claims that “cognitive models of reading often lack [accounts of semantics and discourse]” isn’t quite accurate. I would suggest weakening this claim and citing a few of the discourse-processing models (e.g., Fletcher et al., 1996; Frank et al., 2007; Kintsch, 1998; Myers & O’Brien, 1998; van den Broek et al., 1996).

27. 21/2/2: “7B”  “7 billion”

28. 21/2/4: “124M”  “124 million”

29. 23/1/1: Figure 1 is actually Figure 4.

30. 25/2: In discussing the model’s assumption about predictability, use numbers to list the assumptions; e.g., “…predictability being: (1) graded, …”

31. 26/3/4: Figure 2 is actually Figure 5.

Signed: Erik D. Reichle

Reviewer #2: The paper investigates the suitability of large language models (LLMs) within the framework of a cognitive model of reading, specifically comparing the predictability scores from LLMs (GPT-2 and LLaMA-7B) to those obtained via traditional cloze tasks. The authors demonstrate that LLM-derived predictability scores offer a better fit to human eye-movement data during reading, outperforming cloze-based predictions (as indicated by the Root Mean Square Error (RMSE) between simulated and human eye movements).

The methodology proposed by the authors consists of two "novelties": using LLMs to generate predictability scores and implementing a novel "parallel-graded mechanism" that involves the pre-activation of all predicted words at a given position based on their contextual certainty, which changes dynamically as text processing unfolds. The authors claim that their contribution is the first of its kind in combining a language model with a computational cognitive model of reading, offering a new approach to understanding the interplay between word predictability and eye movements.

Pros:

- The paper introduces an novel integration of LLMs with a cognitive model, contributing to the field of reading research by providing a way of obtaining more dynamic and contextually aware word predictability scores.

- By outperforming traditional cloze methods, the study highlights the potential of LLMs to enhance predictive accuracy in reading simulations.

- The authors are very careful about interpreting the results, ensuring they do not fall into the trap of proposing LLMs as cognitively-plausible models.

Cons/questions:

- The authors only consider the first token of multi-token words, rather than considering the joint probability of all tokens that constitute the word. For example, "customary" is represented as two tokens in LLaMA's vocabulary, and the authors use only p(custom) as the predictability score for the word "customary" instead of p(custom)×p(ary∣custom). I am not sure this is the best approach as it may overestimate the predictability of longer words. Furthermore, this creates discrepancy across different sources of predictability (e.g. with each new tokenizer, the representative part of a multi-token words will differ). The implication of this design choice must be discussed in the paper.

- Although I do not have a problem with it, the choice of GPT-2 and LLaMA should be justified. GPT-2 is not traditionally regarded as an LLM (it is perhaps the last 'small' language model), so it could be presented as a baseline language model to demonstrate the potential benefits of more complex LLMs. It would also be interesting to see whether models with increasing sizes yield better performance or if there is a potential limit to the benefits of LLMs at a certain parameter count.

- The methodology section could benefit from more details. The threshold of 0.01 feels rather arbitrary (how many continuations did you get for 0.005 or 0.001?). It is generally advisable to re-normalize the probabilities after any kind of filtering to ensure they still sum to one. Did you apply softmax again to the probabilities of the remaining words? Having normalized true probability distributions is important for accuracy, especially when comparing different sources.

Overall, this paper makes a contribution to the field by showing how the LLMs can be integrated into the existing cognitive modeling techniques. The use of LLMs to predict word predictability in reading contexts perhaps will lead to more research in reading comprehension.

Reviewer #3: 1. I'm curious whether it's possible to vary the size of the context window of the LLMs as an experimental parameter to see how larger/smaller contexts influence the gaze metrics, specifically in relation to early vs late measures.

2. Page 6. The sentence starting with: "This evidence has led to believe..." seems to be missing a subject (but I'm not a native English speaker so I might be wrong, but it sounds a bit strange to me).

3. I understand it is out of scope for this article but it would be interesting to compare results also across different types of models of eye movements during reading (e.g., SWIFT and EZ Reader). This way one might get a clearer understanding whether LLMs are generally better than Cloze estimates independently of the particular mechanisms of the proposed eye movement model, or alternatively, if the results might depend more on the mechanistic details of the particular model.

4. Page 9. "These results suggest that language models, especially with higher prediction accuracy, are good word predictability..." So there is a clear relationship between an LLMs general prediction accuracy and the fit to eye movement data? Maybe some simple comparison metrics between GPT2 and LLaMa would help the reader along the way here. For the reader it is not clear at this point in the text how these two LLMs differ, for example in terms of vocabulary size or overall prediction accuracy.

5. Page 12, fig 3. Please specify which statistical hypothesis test was used to derive the p-values.

6. Page 14, "the current paper is the first to evidence..." Sounds grammatically strange to my ears.

**Have the authors made all data and (if applicable) computational code underlying the findings in their manuscript fully available?**

Reviewer #1: Yes

Reviewer #2: Yes

Reviewer #3: Yes

PLOS authors have the option to publish the peer review history of their article (what does this mean?). If published, this will include your full peer review and any attached files.

Reviewer #1: **Yes: **Erik D. Reichle

Reviewer #2: No

Reviewer #3: No
---

## [Decision Letter · Decision Letter 1]

9 Sep 2024

Dear Ms. Lopes Rego,

We have now heard back from the three reviewers and are pleased to inform you that your manuscript has been provisionally accepted for publication in PLOS Computational Biology.

Best regards,

Ronald van den Berg

Academic Editor

PLOS Computational Biology

Daniele Marinazzo

Section Editor

PLOS Computational Biology

Reviewer's Responses to Questions

**Comments to the Authors:**

Reviewer #1: Review of manuscript entitled “Language models outperform cloze predictability in a cognitive model of reading” (PCOMPBIOL-D-24-00700R1) by Rego, Snell, and Meeter.

Summary. Because this is a revised version of a manuscript that I previously reviewed, I won’t provide another summary here.

General comments. I was generally positive about the first version of the manuscript, although I did have a handful of substantial concerns and a long list of minor suggestions. The authors did a really nice job of addressing my concerns, as well as tightening up the manuscript. For that reason, I’m going to recommend that the manuscript now be accepted for publication.

Signed: Erik D. Reichle

Reviewer #2: I have reviewed the authors' responses and revisions. While some of the issues I previously raised (e.g., using only the first token to represent words) remain in the pipeline, these concerns are now explicitly discussed in the paper. So any future readers will be aware of these limitations. At this stage, it may be unreasonable to expect the authors to re-run or re-implement the entire pipeline, and I believe the findings and the overall work are interesting for the general audience. I have no further comments.

Reviewer #3: I have re-read the manuscript and I feel that the authors have adequately addressed the points I raised in my review. I recommend that the manuscript be accepted for publication.

**Have the authors made all data and (if applicable) computational code underlying the findings in their manuscript fully available?**

Reviewer #1: Yes

Reviewer #2: Yes

Reviewer #3: Yes

PLOS authors have the option to publish the peer review history of their article (what does this mean?). If published, this will include your full peer review and any attached files.

Reviewer #1: **Yes: **Erik D. Reichle

Reviewer #2: No

Reviewer #3: No

---

## [Editor Report · Acceptance letter]

20 Sep 2024

PCOMPBIOL-D-24-00700R1 

Language models outperform cloze predictability in a cognitive model of reading

Dear Dr Lopes Rego,

I am pleased to inform you that your manuscript has been formally accepted for publication in PLOS Computational Biology. Your manuscript is now with our production department and you will be notified of the publication date in due course.

With kind regards,

Anita Estes
